# *Myo*-D-inositol Trisphosphate Signalling in Oomycetes

**DOI:** 10.3390/microorganisms10112157

**Published:** 2022-10-31

**Authors:** Indu Muraleedharan Nair, Emma Condon, Barbara Doyle Prestwich, John James Mackrill

**Affiliations:** 1Department of Physiology, School of Medicine, University College Cork (UCC), T12 YT20 Cork, Ireland; 2Department of Plant Science, School of Biological, Earth and Environmental Sciences, University College Cork (UCC), T23 TK30 Cork, Ireland

**Keywords:** oomycete, calcium, *myo*-inositol 1,4,5 trisphosphate

## Abstract

Oomycetes are pathogens of plants and animals, which cause billions of dollars of global losses to the agriculture, aquaculture and forestry sectors each year. These organisms superficially resemble fungi, with an archetype being *Phytophthora infestans*, the cause of late blight of tomatoes and potatoes. Comparison of the physiology of oomycetes with that of other organisms, such as plants and animals, may provide new routes to selectively combat these pathogens. In most eukaryotes, *myo*-inositol 1,4,5 trisphosphate is a key second messenger that links extracellular stimuli to increases in cytoplasmic Ca^2+^, to regulate cellular activities. In the work presented in this study, investigation of the molecular components of *myo*-inositol 1,4,5 trisphosphate signaling in oomycetes has unveiled similarities and differences with that in other eukaryotes. Most striking is that several oomycete species lack detectable phosphoinositide-selective phospholipase C homologues, the enzyme family that generates this second messenger, but still possess relatives of *myo*-inositol 1,4,5 trisphosphate-gated Ca^2+^-channels.

## 1. Introduction

Oomycetes are important pathogens worldwide, causing major losses globally [1]. However, they also play a role in nutrient cycling, and have the potential to produce useful chemicals [2,3]. They are generally found in damp terrestrial and freshwater environments, but have also been detected in arid regions such as deserts [4].

Ca^2+^ is a key second messenger utilized by all domains of life. Due to its high abundance in nature and potential cytotoxic effects, cells have evolved to maintain unstimulated cytoplasmic Ca^2+^ concentrations at sub-micromolar concentrations, compared to low millimolar extracellular levels [5]. Changes in cytoplasmic Ca^2+^ concentrations, due to the concerted actions of channels, transporters and buffers, control a wide array of cellular processes [6,7]. High cytoplasmic levels of Ca^2+^ can be lethal to cells [8]. Therefore, to avoid excessive Ca^2+^ accumulation, organisms have evolved mechanisms to expel excess Ca^2+^ and to store it in organelles such as the endoplasmic reticulum, vacuoles, mitochondria and the endolysosomal system [5,7,9,10]. In turn, this has facilitated the use of Ca^2+^ as a second messenger to regulate cell biology, by Ca^2+^ fluxes across the membranes of the cell-surface and intracellular stores.

Ca^2+^ release from intracellular stores can be rapidly triggered by a wide array of stimuli, allowing cells to produce fast responses [11]. Ca^2+^ channels located in the cell-surface membrane can be triggered to open by an array of stimuli, including mechanical forces, neurotransmitters, hormones, nutrients, membrane depolarization, or changes in temperature, allowing extracellular Ca^2+^ to flow into the cell [6,12]. However, for extracellular cues to activate Ca^2+^ channels in the membranes of intracellular Ca^2+^ stores, second messengers must be employed. There are several second messengers that elicit Ca^2+^ release from these intracellular stores, including *myo*-D-inositol 1,4,5-trisphosphate (IP_3_), cyclic ADP ribose, and nicotinic acid dinucleotide phosphate (NAADP) [7].

As the endoplasmic reticulum is a major Ca^2+^ store within most eukaryotic cells, its membrane is an important site for the action of Ca^2+^ release channels, including the IP_3_ receptors (IP_3_Rs) and ryanodine receptors (RyRs) [13]. In addition to the endoplasmic reticulum, plants and fungi also accumulate Ca^2+^ in vacuoles and other organelles. Paradoxically, structural homologues of mammalian IP_3_Rs have not been detected in the genomes of late-branching plant or late-branching fungal taxa [14]. However, they are encoded by the genomes of early-branching groups such as *Volvox* spp. [15] and *Mucor* spp [16]. In addition, whilst concentrated in the endoplasmic reticulum, IP_3_Rs and RyRs have also been reported to be present in the cell surface membrane, the sarcoplasmic reticulum, mitochondria, the endolysosomal system, the Golgi and the nuclear envelope [6,12].

In animal cells, depletion of intracellular Ca^2+^ stores, which can occur as a consequence of opening of IP_3_Rs, causes conformational changes in stromal interaction molecules (STIM): proteins that sense endoplasmic reticulum intraluminal Ca^2+^ concentrations. On sensing Ca^2+^-depleted stores, STIM can interact with Orai Ca^2+^ channels in the cell-surface membrane, thereby facilitating store-operated Ca^2+^ entry [17]. Another way by which Ca^2+^ can enter eukaryotic cells is via Na^+^/Ca^2+^-exchangers (NCX). These are surface membrane transporters that use the entry of three Na^+^ to expel one Ca^2+^. Under certain conditions, such as high intracellular Na^+^ or in depolarized cells, NCX can operate in reverse mode, driving Ca^2+^-entry [18]. Elevation of cytoplasmic Ca^2+^ via by any mechanism (entry or release) can be amplified by RyRs- or IP_3_R-dependent Ca^2+^-release, in a process termed Ca^2+^-induced Ca^2+^-release (CICR) [11].

Ca^2+^ plays a key role in controlling oomycete biology [1,9]. Many eukaryotic Ca^2+^ signalling components are conserved in oomycetes; however, several channels and receptor families are either apparently absent or possess distinctive structures. For example, homologues of Orai and STIM, essential components of store-operated Ca^2+^-entry in other eukaryotes, could not be detected in the oomycetes *Phytophthora infestans* and *Saprolegnia declina.* However, these oomycetes possess homologues of mammalian IP_3_Rs, whose opening would lead to Ca^2+^-store depletion [9,19].

G-protein-coupled receptors (GPCRs) and receptor tyrosine tyrosine kinases (RTK) are located in the surface membrane where they can initiate Ca^2+^ signalling in response to stimuli, such as nutrients, hormones and growth factors. Once activated, these receptors couple to enhanced phosphoinositide-selective phospholipase C (PI-PLC) activity. Such GPCRs are associated with heterotrimeric G-protein complexes, consisting of α, β and γ subunits. On ligand-receptor binding, the subunits dissociate and can then interact with transducer proteins. G-protein subunits of the α_q_ or α_11_ subtypes can stimulate certain PI-PLC isozymes (PI-PLC β1–β3 in mammals) [20]. Ligand-activated RTKs have intrinsic kinase activity and initiate signalling by phosphorylating distinct PI-PLC isozymes (γ1 and γ2) at key tyrosine residues. Other PI-PLC families (δ, ε, η, ζ) are activated by various stimuli, including heterotrimeric G-protein βγ subunits (for η), monomeric G-proteins (such as Ras and Rho, for ε) and Ca^2+^ itself (for δ, ε, η, ζ) [21]. Active PI-PLCs hydrolyse a minor membrane phospholipid, phosphatidylinositol 4,5–biphosphate (PIP_2_), to generate the second messengers 1,2-diacylglycerol (DAG) and IP_3_, Figure 1. DAG is membrane associated lipid, which exerts its effects by stimulation of protein kinase C isozymes which phosphorylate target proteins, modifying their function, thereby regulating cellular activities [22].

In contrast to DAG, IP_3_ is a soluble second messenger which diffuses through the cytoplasm, to interact with IP_3_ receptors (IP_3_R) located in the membranes of intracellular Ca^2+^ stores, such as the endoplasmic reticulum. Binding of four molecules of IP_3_ per tetrameric IP_3_R promotes a channel-open conformational state of this complex, allowing Ca^2+^ release [23]. In addition, both IP_3_Rs and RyRs display biphasic relationships between channel gating and both cytoplasmic and intraluminal Ca^2+^ concentrations [11]. Preliminary analyses of oomycete conceptual proteomes indicates that IP_3_ signalling in oomycetes is distinct from that in other eukaryotes, such as plants and animals [4,9,19,24]. For example, the only candidate proteins in oomycetes sharing high identity with RyRs do so by homology with two “RyR-(R) domains”. These homologues are more closely related to the polycystic kidney disease (PKD) member of the transient receptor potential channel superfamily than they are to RyRs. The presence of these oomycete-specific PKDRR proteins in *P. infestans* has been verified using Western immunoblotting and immunofluorescent microscopy approaches [19]. Such differences between signalling molecules may provide a route for development of new strategies for combatting oomycete pathogens and consequently, are the focus of this review.

## 2. Oomycetes

Due to their similar morphologies, growth patterns and life cycles, oomycetes were long classified amongst fungi [4,25,26,27]. However, despite their superficial similarities, the ancestors of oomycetes and fungi diverged a long time ago and are found within different branches of the eukaryotic phylogenetic tree. From molecular clock models, oomycetes are estimated to have diverged from their last common ancestor with other eukaryotes about 430 to 400 million years ago. The two major orders, Peronosporales and Saprolegnialea, are estimated to have diverged from each other 225 to 190 million years ago [28]. Taxonomically placed into the *Straminipilia* kingdom, oomycetes are closely related to diatoms and brown algae [4]. As such, they are thought to have originated in marine environments, moving into freshwater and terrestrial environments as their hosts evolved [29].

Oomycete genomes (50–250 Mb) are much larger than those of fungi (10–40 Mb) [1]. Another main distinction between oomycetes and fungi are the components of their cell walls. In fungi, chitin is the main component, whereas in oomycetes there is very limited chitin, with the walls being made of mostly cellulose and other β-glucans [27,30].

Due to their evolutionary distance from ‘true fungi’, oomycetes utilize distinct genetic and biochemical mechanisms to infect their hosts. Genome sequences of important oomycete pathogens have also highlighted biochemical disparities between oomycetes and other eukaryotes [3,24]. Horizontal gene transfer between species and ‘host jumping’ are also thought to play a role in oomycete biochemical pathways, influencing pathogenicity, signal transduction and motility [2,4]. Within oomycetes, there are a wide variety of lifecycles, from biotrophy (obtaining nutrients from living hosts) to necrotrophy (obtaining them from dead tissues). Many pathogenic oomycetes, including *Phytophthora* spp., are hemi-biotrophic (switching from biotrophy to necrotrophy during disease progression) [2,4].

Oomycetes are the cause of many devastating plant and animal diseases globally, often leading to major losses of crops, trees and fish-stocks [26,31]. One example is *Phytophthora infestans*, which causes late blight in tomato and potato crops. This pathogen is well known as the agent that caused widespread devastation of the potato crop in Ireland and across Europe in the mid-19th century leading to the Great Irish Famine [29]. Oomycete pathogens, are notoriously difficult to control and can affect a wide range of hosts and ecosystems. Economic losses are not only incurred by crop failure, but also those involved in the application of pesticides against oomycetes. For *P. infestans* alone, global costs are estimated at about 10 billion US dollars per year [1]. Many anti-oomycete pesticides are becoming increasingly ineffective, due to development of resistance in oomycete populations, or the banning of these chemicals due to environmental and health concerns [32].

One of the most important factors in oomycete pathogenicity is the ability to rapidly produce zoospores, generated within structures called sporangia. Zoospores are released from sporangia during cold and wet conditions. These mobile spores then locate a new host, encyst, and germinate to infect the host via haustoria, instigating the infection cycle once again [27]. Ca^2+^ signalling plays an important role in regulating this aspect of oomycete infection. Therefore, understanding the underlying Ca^2+^ signalling mechanisms at play could allow for the infection to be targeted without harming the host [9].

However, despite the importance of these pathogens, knowledge pertaining to the mechanisms involved in controlling their cell biology is lacking. Understanding their unique signalling mechanisms may provide insight into new control methods [3,19]. As a result of their distinct biochemistry, classical fungicides have long proven ineffective against oomycetes. For example, oomycetes do not synthesise ergosterol making them resistant to the effects of azole fungicides [33]. Another example, Metalaxyl, interferes with RNA polymerases and has proven to be highly effective against oomycetes. This has a wide spectrum of activity, however, resistance in *P. infestans* was reported within a year of its initial use and is now widespread in certain subpopulations of this pathogen [34]. Thus, finding new ways to target these pathogens is imperative. A thorough understanding of oomycete physiology and infection mechanisms could open up a wide variety of possible control strategies using both chemical and biocontrol approaches. 

## 3. Ca^2+^ and Phosphoinositide Signalling in Oomycetes

### 3.1. Oomycete Ca^2+^ Signalling

Like other eukaryotes, a large electrochemical gradient for Ca^2+^ exists across the cell membrane of oomycetes, with cytoplasmic levels of about 100 nM in *Phytophthora cinnamomi* [35], relative to the high micromolar to low millimolar concentrations in the extracellular environment. Changes in cytoplasmic Ca^2+^ concentration have been reported to regulate multiple aspects of oomycete biology [9]. For example, stretch-activated Ca^2+^ channels towards the hyphal tips in the oomycete *Saprolegnia ferax* act as a source of Ca^2+^ for sensing hyphal expansion [36]. Cold-shock is a stimulus that triggers zoosporogenesis and zoospore release in many *Phytophthora* spp. [1,37]. Cytokinesis and zoospore formation in *P. cinnamomi* sporangia is triggered by a cold stimulus is associated with a rise in cytoplasmic Ca^2+^ [36]. Oomycete zoospores migrate towards favourable, or away from unfavourable, chemical stimuli in a process termed chemotaxis. Silencing of the Gα subunit-encoding gene (potentially involved IP_3_ -mediated Ca^2+^ signalling) in *P. sojae* altered the chemotaxis of zoospores towards the soybean isoflavanone, daidzein [38]. The motility of *Pythium porphyrae* (cause of root-rot of the seaweed *Porphyra yezoensis*) zoospores is inhibited by extracellular Ca^2+^, whereas germination of its cysts is blocked by ethylene glycol-bis(2-aminoethylether)-N, N, N, N -tetraacetic acid (EGTA), a Ca^2+^ chelator [39].

### 3.2. Phosphoinositide and Myo-D-inositol 1,4,5-Trisphosphate (IP_3_) Signalling

IP_3_ is a widely-utilized second messenger found in most eukaryote systems [40]. IP_3_Rs and RyRs form a superfamily of tetrameric channels that release Ca^2+^ from intracellular stores [6,16]. The binding of signalling molecules (IP_3_, Ca^2+^, cADPR, possibly NAADP) or allosteric interactions with other proteins can activate these channels allowing release of Ca^2+^ [7].

In oomycetes, putative IP_3_-mediated Ca^2+^ signalling has been well studied in relation to zoosporogenesis [41]. The production and encystment of zoospores plays a key role in the pathogenicity of many oomycetes [25]. A short period of cold shock is required for zoospores to be released from sporangia. This cold shock reduces fluidity of the cell surface membrane, resulting in stimulation of membrane bound sensors such as mechano-sensitive cation channels and GPCRs [1,37]. In other eukaryotes, activation of GPCRs can initiate the PLC–IP_3_-Ca^2+^ release mechanism, whereas mechano-channel activation leads to Ca^2+^ -influx that can be amplified by CICR, with either mechanism potentially resulting in Ca^2+^ release from the endoplasmic reticulum or vacuoles via IP_3_Rs. Despite a clear role for Ca^2+^ in triggering zoospore encystment, PI-PLC homologues have yet to be detected in *Phytophthora* spp. [24]. However, pharmacological studies suggest that PLC–IP_3_-dependent signalling plays a role in *P. infestans* zoosporogenesis. Both the PLC-inhibitor U73122 and the IP_3_-antagonist 2-aminoethoxydiphenyl borate (2-APB) blocked zoosporogenesis in *P. infestans* [41]. Of a family of seven nuclear LIM interactor-interacting transcription factors, four members were upregulated in zoospores, with this expression being inhibited by either U73122 or 2-APB [42]. The properties of some of the chemicals used to study oomycete PI signalling are summarized in Table 1. A key point to note is that most of these small molecules exert effects on multiple targets in mammalian cells, bringing their specificity of effect in oomycetes into doubt.

In the current study, the presence of homologues of *Homo sapiens* PI signalling proteins was investigated in the oomycete orders Peronosporales, Saprolegniales and Albuginales. Animals, fungi, and plants were also examined, for comparative purposes. Query sequences were: Gα_q_, PI-PLCs (α, β, δ, ε, η and ζ), IP_3_R channels (ITPR), IP kinases (including ITPK1, IPPK, ITPKA and IPMK) and IP phosphatases (including IMPA1, INPP1, INPP4, INPP5 and MINPP1). Findings from this study are summarized in Figure 2 and are represented in full detail in Appendix A.

Homologues of *Homo sapiens* IP signalling proteins were investigated in *Caenorhabditis elegans* (Metazoa, Nematoda), *Neurospora crassa* (Fungi, Sordariales), *Arabidopsis thaliana* (Embryophyta, Bassicales), *Phytophthora infestans* (Oomycota, Peronosporales), *Plasmopara halstedii* (Oomycota, Peronosporales), *Pythium oligandrum* (Oomycota, Peronosporales), *Saprolegnia parasitica* (Oomycota, Saprolegniales), *Pythium oligandrum* (Oomycota, Saprolegniales) and *Albugo laibaichii* (Oomycota, Albuginales), using the Protein BLAST tool at NCBI (https://blast.ncbi.nlm.nih.gov/Blast.cgi?PAGE=Proteins). The query genes were *H. sapiens* GNAQ, PI-PLCs (only δ1 shown), ITPR, IP-kinases (ITPK1, IPPK, ITPKA, IPMK), and IP-phosphatases (IMPA1, INPP1, INPP4, INPP5 and MINPP1). The look-up-table (LUT, upper left) indicate estimates of the probability that homology between sequences occurred by chance. The cut-off for this value used is 1 x10^−4^. ‘ND’ indicates no detectable homology. ‘C2’ indicates limited homology, due to a common C2 domain. Further details are available in Appendix A.

### 3.3. Oomycete GPCRs, G-Proteins and RTKs

The genomic sequences of *P. sojae* and *P. ramorum* indicate the presence of a single Gα and a single Gβ subunit in *Phytophthora* spp. [44]. However, there are 24 hypothetical GPCR proteins in these species, 12 of which are distinct from the GPCRs other eukaryotes in that they contain a C-terminal phosphophatidylinositol phosphate kinase (PIPK) domain. In *P. infestans*, silencing of one gene encoding a GPCR-PIPK (termed GK4) inhibited sporangial development and virulence [45]. In *P. sojae*, silencing of either GK4 or GK5 impaired pathogenicity, but only deletion of GK4 promoted encystment and reduced cyst germination [46]. Consequently, these GPCR-PIPKs have important roles in oomycete biology and might represent a means of directly coupling extracellular signals to increased phosphorylation of PI lipids, bypassingPLC activation [47].

In *P. sojae*, a putative GPCR called GPR11 is upregulated in zoospores. Silencing of this gene impaired zoospore release, encystment and germination, as well as pathogenicity. However, interaction between GPR11 and Gα could not be demonstrated using the yeast two-hybrid complementation system, nor did silencing of these genes have related effects on the expression of specific target genes [48].

G-protein mediated Ca^2+^ signalling potentially plays an important role in zoosporogenesis and cyst formation in *P. infestans*, since Gα (*pigpa1*) and Gβ (*pigpb1*) genes are upregulated during these developmental stages [49]. Silencing of the sole Gα subunit in *P. infestans* lead to aberrant swimming patterns in zoospores [43]. Similarly, in *P. sojae*, the Gα subunit is essential for chemotaxis towards soybean isoflavones [38]. It also plays a role in locating sites for penetration of host tissues and forms part of a signalling pathway with a regulator of G-protein signalling (RGS) proteins, a GTP-hydrolysing protein that can inactivate the Gα subunit [50]. In contrast, silencing of the Gβ subunit in *P. infestans* inhibited vegetative growth and formation of sporangia [51]. These experiments indicate that in oomycetes, different subunits of heterotrimeric G-proteins initiate separate signalling events, with distinct cellular outcomes. There is no evidence that homologues of mammalian heterotrimeric G-protein γ-subunits are encoded within the genomes of oomycetes [24,44].

Mastoparan is a peptide obtained from the venom of the wasp *Vespula lewisii*. In addition to membranolytic properties, mastoparan is a direct activator of heterotrimeric G-proteins, which mimics activated GPCRs [52]. In many eukaryotic cells, including those from animals and plants, application of mastoparan promotes PIP_2_ hydrolysis, accumulation of IP_3_ and DAG [53,54]. In contrast, mastoparan appears to stimulate phospholipase D (PLD) in *P. infestans*, leading to an increase in phosphatidic acid (PA) levels, with no evidence of increased PIP_2_ hydrolysis [21,43,49,55]. This suggests that the signalling mechanisms elicited by mastoparan in *P. infestans* are distinctive from those in other eukaryotes.

In metazoans (animals), the IP_3_ signalling pathway can also be activated through RTKs, which phosphorylate members of the PLCγ family to drive PIP_2_ hydrolysis. Although candidate tyrosine kinases have been identified in the genome of *P. infestans* [56], their relationships to metazoan RTKs and potential roles in Ca^2+^ signalling have not been established to date.

### 3.4. Oomycete Phosphoinositide-Selective Phospholipase C (PI-PLC) Isozymes

PI-PLCs hydrolyse the membrane phospholipid PIP_2_ to generate IP_3_ and DAG. IP_3_ can be inactivated (in terms of its ability to effectively gate IP_3_Rs) by progressive phosphorylation to generate higher order inositol polyphosphates. It can also be dephosphorylated to produce inositol, which can be used to regenerate PIP_2_ [57,58,59].

Substantial quantities of the PI-PLC substrate PIP_2_ could not be detected in *P. infestans*, preventing further analysis of its biological roles [55]. Furthermore, genomewide analysis of *P. infestans* failed to identify any PI-PLC homologues. However, PLD, DAG kinase, and phosphatidylinositol phosphate kinase (PIPK) encoding genes were detected in this oomycete [24]. In contrast, PIP_2_ has been unequivocally detected in the hyphae of *P. cinnamomi,* using liquid chromatography-mass spectroscopy approaches [60]. Chilling-induced zoosporogenesis of *P. infestans* (a Ca^2+^-dependent process) did not involve convincing IP_3_ accumulation [61]. In contrast, substantial IP_3_ accumulation was detected during pectin-stimulation of *P. palmivora*, with this plant-derived stimulus promoting differentiation of zoospores into cysts. However, the change in IP_3_ concentration was not rapid enough, nor was of large enough magnitude, to be the driver of zoospore differentiation. Furthermore, it has been estimated that the unstimulated cytosolic concentration of IP_3_ in *P. palmivora* zoospores was in the order of 5 μM [62], about two orders of magnitude greater than that in metazoan cells [63].

In the current study, searching for homologues of human PI-PLC proteins encoded by genomes from the Stramenopile-Alveolate-Rhizaria (SAR) group failed to detect these enzymes in *P. infestans*, Figure 2 and Figure 3. However, other oomycetes possess unambiguous homologues of PI-PLCs. These generally have a predicted protein domain structure consisting of a PLC-catalytic domain close to the C-terminus, in combination with an EF-hand Ca^2+^-binding site closer to the N-terminus, Figure 3 and Appendix A. Other oomycete PI-PLCs contain an additional candidate pleckstrin homology (PH) domain, N-terminal to the EF-hand domain. These predicted protein architectures are reminiscent of mammalian PI-PLCε isozymes, which are activated by increased cytoplasmic Ca^2+^ concentrations, suggesting that oomycete PI-PLCs of this type may serve to amplify transient Ca^2+^ signals [64]. Two distinctive types of oomycete PI-PLCs form phylogenetically distinct clusters. These types were only found in the genome of *P. cactorum*. One contains a PLC domain only; whereas the other bears an additional Ca^2+^-dependent lipid-binding, or C2-domain, Figure 3.

### 3.5. Phosphatidic Acid in Oomycetes

Phosphatidic acid (PA) is a minor structural component of eukaryotic membranes. It is an important second messenger in many organisms including plants, animals and oomycetes, through its abilities to interact with a range of proteins and to induce membrane curvature [65]. PA is involved in cyst production in both *P. palmivora* [62] and in *P. infestans* [55,62]. PA can be generated by hydrolysis of phosphatidylcholine by phospholipase D (PLD) isozymes [66]. The other product of PLD activity, choline, is reported to bind to sigma-1-receptors in the endoplasmic reticulum, sensitizing IP_3_R gating to IP_3_ [67]. An alternative pathway for formation of PA is through phosphorylation of DAG by DAG kinases [68]. PA can also be produced by a third pathway, requiring the action of lysophosphatidic acid acyltransferases on lysophosphatidic acid [65].

Both the PLD- and PLC-dependent PA production pathways are activated by mastoparan in plant and *P. infestans* cells [55]. In *P. palmivora*, pectin induces a large and rapid accumulation of PA, which are consistent with it playing a second messenger role in zoospore encystment [62]. It is hypothesized that in *Phytophthora* spp., the majority of PA production occurs via the PLD pathway. This could explain the large number of PLDs, the limited range of DAG kinases, and the apparent lack of PLC genes in *P. infestans* [24].

### 3.6. IP_3_Rs in Oomycetes

IP_3_Rs have been studied across a wide variety of eukaryotes due to their role in Ca^2+^ signalling. However, their functions within oomycetes are yet to be elucidated. Different oomycete species appear to possess different numbers of IP_3_R homologues. For example, *P. infestans* encodes one putative IP_3_R homologue, whereas the *Saprolengnia diclina* genome contains 4 candidate homologues [9,19].

Vertebrate genomes encode three closely related IP_3_R subunits (IP_3_R1–3, sharing 60–80% amino acid identity), which assemble into homo and heterotetrametric channel complexes. The size of the subunits, at approximately 2700 amino acid residues, make them among the largest ion channel complexes identified to date. The three IP_3_R subtypes differ in their subcellular distributions, affinities for IP_3_ (IP_3_R2 > IP_3_R1 > IP_3_R3) and modulation by associated proteins [72]. Even though in mammals, most cells express two or three different IP_3_R subtypes, they differ in their distribution between tissues and subcellular domains.

Mammalian IP_3_Rs are activated by the binding of IP_3_ to a clam-like structure, the IP_3_-binding core (IBC), located near the N-terminus of each IP_3_R subunit. Binding of IP_3_ alters the interaction between the IBC and the N-terminal suppressor domain (SD), Figure 4. These changes are proposed to disrupt interactions between the N-terminal regions of the four subunits of the IP_3_R, leading to gating of the intrinsic ion channel. The channel pore is formed by transmembrane domains towards the C-terminus of each IP_3_R subunit [73]. Binding of the IP_3_ to the receptor evokes conformational changes that expose of an activating Ca^2+^ binding site, leading to channel gating. There is also an inhibitory Ca^2+^ binding site, which promotes channel closure at higher Ca^2+^ concentrations [74].

Putative oomycete IP_3_R homologues were identified in conceptual translations of oomycete genomes, using the basic local alignment of sequences (BLAST) tool with human type 1 IP_3_R (ITPR1) protein as the query sequence [19]. Oomycete IP_3_Rs show poor conservation of amino acids within the IBC that are critical for IP_3_-binding to mammalian IP_3_Rs, Figure 4. This suggests that either IP_3_ is not an endogenous ligand of oomycete IP_3_Rs, or that these channels interact with the second messenger in a different way. We consider that the latter possibility is more likely, since at least some oomycetes display stimulus-dependent increases in IP_3_. This concept is not without precedent, since trypanosome IP_3_Rs display poor conservation of critical residues in the IBC, but have been experimentally demonstrated to act as IP_3_-activated Ca^2+^-release channels [75,76]. Furthermore, several oomycete Orders possess candidate PI-PLCs. Finally, all oomycete genomes analysed encode enzymes with the potential for the inactivation of IP_3_.

## 4. Enzymes That Inactivate IP_3_

IP_3_ is metabolized into forms less effective at IP_3_R gating, by removal of phosphate groups by inositol polyphosphate (IP) phosphatases and by phosphorylation by IPkinases. IP_3_ can be progressively dephosphorylated to inositol. DAG is phosphorylated by DAG-kinase-ɛ to form phosphatidic acid (PA). Additional enzymatic steps result in the formation of PIP_2_ from these products [57], Figure 1. Initially, IP_3_ is metabolized in two ways; either dephosphorylation by a 5-phosphatase to give inositol 1,4-bisphosphate (IP_2_) or through phosphorylation by IP_3_K to produce IP_4_ [58].

### 4.1. Oomycete IP Kinases

Metabolism of IP_3_ results in the production of diverse inositol polyphosphates (IPs), such as IP_4_, IP_5_, IP_6_, and PP-IP_5_ [59]. In mammals, inositol 1,4,5-trisphosphate 3-kinase (IP_3_ 3-kinase/IP3K) plays a key role in maintaining Ca^2+^ homeostasis. It phosphorylates IP_3_ to inositol 1,3,4,5-tetrakisphosphate (IP_4_). IP_4_ is another second messenger, which inhibits store-operated Ca^2+^-entry; reduces translocation of Akt, Bkt and Itk protein kinases; remodels vesicular transport; regulates GTPase-activating proteins; and possibly mobilizes intracellular Ca^2+^ by acting synergistically with IP_3_ [77].

IP_3_ 3-kinases (ITP3Ks) are a family of enzymes that phosphorylate IP_3_ to IP_4_ [59]. There are two major functional domains in mammalian ITP3Ks: a highly conserved C-terminal catalytic domain and a divergent N-terminal regulatory domain. The catalytic core of mammalian ITP3Ks consists of two domains: a large α/β-class structure and a small α-helical structure [78,79]. Three ITP3Ks are encoded by the human genome: ITPKA, ITPKB, and ITPKC. All share a conserved C-terminal catalytic domain but differ in mechanisms of regulation, as well as in their tissue distribution [80].

Mammalian ITP3Ks can be activated by calmodulin (CaM) in a Ca^2+^- dependent manner. CaM recognizes sequences which contain amphiphilic α-helices with clusters of positively charged and hydrophobic amino acids [81]. ITPKs are stimulated directly by Ca^2+^/calmodulin binding [82]. The mechanism by which this works is that CaM recognizes sequences that contain amphiphilic alpha-helices with clusters of positively charged and hydrophobic amino acids [83]. Certain sequences are required for CaM binding and enzyme activation and this level of stimulation by this interaction appears to be specific to cell, tissue, or isoform. ITP3Ks from nematodes and *Arabidopsis thaliana* lack the CaM-binding sites and therefore are insensitive to Ca^2+^ and CaM [80]. 

All ITPKs belong to a larger structural family, the inositol polyphosphate kinases or IPKs. Inositol phosphate multikinase (IPMK) is widely distributed in the kingdoms of animal, plant, and yeast [84]. Cellular IP_3_ serves as a substrate for both ITP3K and IPMK to form IP_6_. IPMK, but not ITP3K, is the major enzyme in IP_6_ synthesis. ITPK mainly functions in IP_4_ synthesis from IP_3_ [85]. Other kinases generate higher IPs. These higher IPs have multiple regulatory roles in cells, including control of DNA repair, endocytosis and chloride channel gating [58,59]. Inositol tetrakisphosphate 1-kinase (ITPK1) phosphorylates position 1 of Ins(3,4,5,6)P_4_ to generate Ins(1,3,4,5,6)P_5,_ or the 5 position of Ins(1,3,4)P_3_ to produce Ins(1,3,4,5)P_4_ [86]. Inositol-pentakisphosphate 2-kinases (IPPK) phosphorylate Ins(1,3,4,5,6)P_5_ at the 2 position to generate Ins(1,2,3,4,5,6)P_6_ [87].

Among the oomycetes investigated, ITP3K homologues were only detected in *Pythium oligandrum* and in *Saprolegnia parasitica*. These homologues only display weak homology with *H. sapiens* ITPKA and display greater identity to metazoan IMPK proteins, meaning that they are more likely to belong to the latter class of enzymes, Figure 2 and Appendix A. In contrast, well-supported homologues of IMPK were detected in all translated oomycete genomes investigated, suggesting that these multikinases are likely to operate in these organisms. Most oomycetes examined also possess ITPK1 and IPPK homologues, suggesting that they have the molecular apparatus for both inactivating the Ca^2+^-releasing second messenger IP_3_, and for generating higher IPs that have their own cellular roles.

### 4.2. Oomycete IP Phosphatases

The Ca^2+^-mobilizing activity of IP_3_ can also be ablated by IP phosphatases. Some IP phosphatases only dephosphorylate lipid-associated inositol groups (PI, PIP_2_, PIP_3_), some only soluble inositol phosphates, while others have broad substrate specificities [57]. Inositol monophosphatases (IMPAs) convert IP_1_ to inositol, a key step in the generation of PI lipids. IMPA can also dephosphorylate a wide range of substrates, including Ins(1,3)P_2_, Ins(1,4)P_2_, glucose-1-phosphate, fructose-1-phosphate and β-glycerol phosphate [88].

Inositol polyphosphate 1-phosphatases (INPP1) catalyses the removal of the 1-phosphate from Ins(1,4)P_2_ and Ins(1,3,4)P_2_. Both IMPA and INPP1 enzymes are inhibited by lithium, an ion that promotes IP_3_ accumulation and inhibits inositol formation within cells [89]. Inositol polyphosphate 4-phosphatase type I isozymes (INPP4A and INPP4B) remove the phosphate from the 4 position of the lipids PI(3,4)P_2_ and PI(1,4)P_2_, in addition to those of the soluble Ins(1,4)P_2_ and Ins(1,3,4)P_3_ [90]. Inositol polyphosphate-5-phosphatases are a multigene family of signalling enzymes. Of these, INPP5A, also called the SH2-containing inositol phosphatase (SHIP), dephosphorylates Ins(1,4,5)P_3_, Ins(1,3,4,5)P_4_ and PI(3,4,5)P_3_ at the 5 position, thereby inactivating these signalling molecules [91]. Multiple inositol polyphosphate phosphatase (MINPP1) is a member of the histidine phosphatase family of proteins, which removes phosphate groups from a wide variety of substrates, including IP_3_, IP_4_, IP_5_ and IP_6_. MINPP1 is highly conserved among eukaryotes [92]. It also serves as a signalling hub, in the interconversion of different forms of IP-containing molecules [93].

The current study provides strong evidence for homologues of IMPA1, INPP4 and MINPP1 in all oomycetes investigated, Figure 2. In contrast, no robust homologues of INPP5 were detected and matches obtained for INPP1 were dubious. For example, the hit retrieved from *P. infestans* with a *H. sapiens* INPP1 query sequence is annotated as an ecdysteroid kinase and probably weakly matches mammalian INPP1 through a shared nucleotidase domain. Overall, these findings indicate that oomycetes display both similarities and differences with other eukaryotes, in terms of their mechanisms for dephosphorylating IPs.

## 5. Perspectives

Oomycetes are economically important pathogens of plants and aquatic animals. The current work and previous studies have highlighted key differences in PI signalling between oomycetes and their hosts. while most oomycete genomes investigated appear to encode IP_3_R homologues, only those of saprolegnians contain robust PI-PLC homologues. Oomycete IP_3_R homologues display limited conservation of key residues required for IP_3_ binding by their mammalian counterparts. Furthermore, oomycete genomes encode an extensive suite of enzymes for IP_3_ metabolism: at the very least homologues of the IP kinases ITPK1, IPPK, and IPMK; and of the IP phosphatases IMPA1, INPP4 and MINPP1. Given these observations and conflicting findings on detection of PIP_2_ and IP_3_ in oomycetes, it is likely that IP_3_-signalling in these pathogens is very different from that in other eukaryotic groups. Potential mechanisms of coupling between extracellular stimuli and IP_3_Rs in oomycetes include amplification of Ca^2+^-influx by CICR or activation of Ca^2+^-dependent PI-PLCs (in some taxa); or PLD-dependent pathways, such as sensitization of IP_3_Rs via choline binding to sigma-1-receptors. Mechanisms of IP_3_R-mediated Ca^2+^-signalling that are distinct from those in mammals have been observed in other eukaryotes. For example, the IP_3_R of the euglenozoan *Trypanosoma brucei* is located in a specialized organelle called the acidocalciosome, where it is activated by products of intraluminal polyphosphate hydrolysis [76].

Key experimental steps in the further characterization of oomycete PI-signalling are: (1) lipidomic analyses of the turnover of signalling lipids in response to extracellular stimulation. For example, do mastoparan, cold-shock or cues from hosts stimulate the conversion of PIP_2_ to PIP? (2) Determination if oomycete IP_3_Rs can act as IP_3_-gated Ca^2+^-channels. This might be achieved in heterologous expression of these proteins. If these proteins form channels that are not gated by IP_3_, which second messenger(s) are they opened by? (3) Are oomycete IP-kinases and -phosphatases functional? If so, what are their biological roles and how do they differ from their counterparts in other eukaryotes?

Addressing these questions is likely to lead to new insights into oomycete biology and to novel strategies for combatting these pathogenic organisms.

## Figures and Tables

**Figure 1 microorganisms-10-02157-f001:**
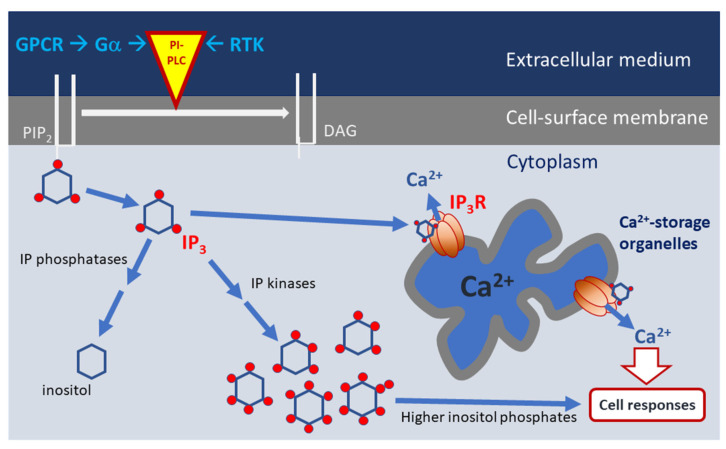
Generalized scheme of phosphoinositide signalling in eukaryotic cells. Extracellular signals ca interact with cell-surface receptors, such as 7-transmembrane G-protein coupled receptors (GPCRs) or receptor tyrosine kinases (RTKs). GPCRs are associated with heterotrimeric GTP-binding proteins. The Ga_q/11_ subunits of these dissociate from activated receptors to stimulate b-subtypes of phosphosphingolipids-specific phospholipase C (PI-PLC). Activated RTKs stimulate g-subtypes of PI-PLC, by phosphorylating specific tyrosine residues. PI-PLCs cleave the minor membrane lipid phosphatidyl 4,5-bisphosphate (PIP_2_) to form soluble IP_3_ and membrane associated DAG. IP_3_ interacts with IP_3_Rs located in Ca^2+^-storage organelles such as the endoplasmic reticulum and vacuole, gating an intrinsic ion channel to release Ca^2+^ into the cytoplasm. IP_3_ can be inactivated by progressive dephosphorylation by inositol phosphate (IP) phosphatases, to generate inositol. This is recycled during the regeneration of PIP_2_. IP_3_ is also phosphorylated by IP kinases to form higher IPs, which have diverse biological functions distinct from the Ca^2+^-releasing role of IP_3_.

**Figure 2 microorganisms-10-02157-f002:**
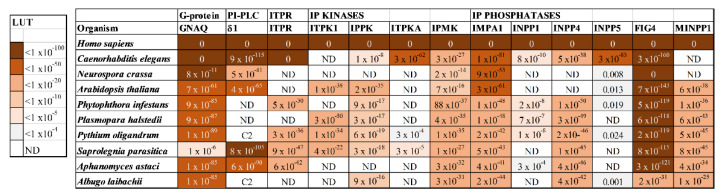
Oomycete genomes encode putative Gαq/11 subunits, IP kinases and IP phosphatases, but only members of the Order Saprolegniales possess detectable PI-PLC homologues.

**Figure 3 microorganisms-10-02157-f003:**
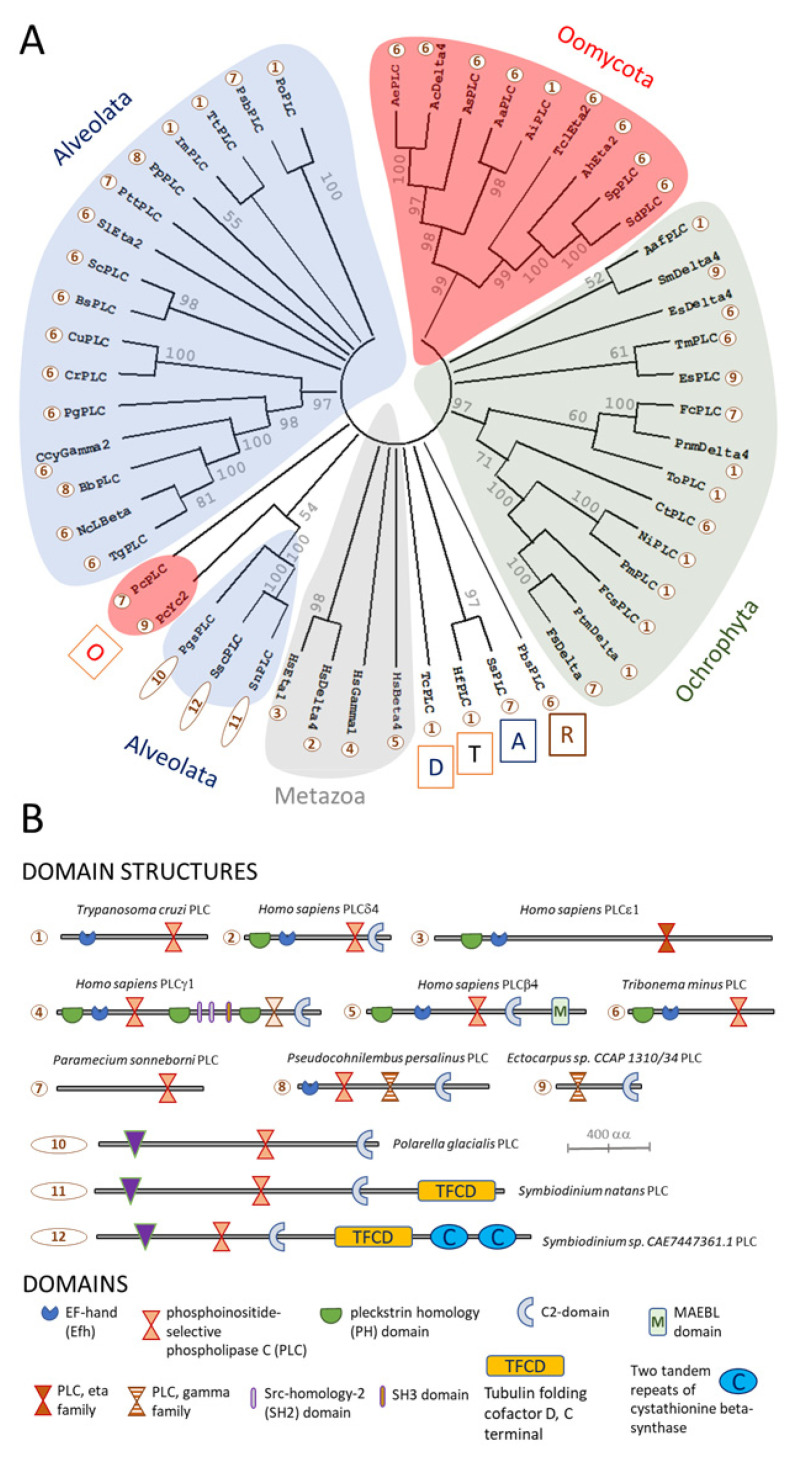
Phylogeny and protein domain architectures of PI-PLC proteins from members of the Stramenopile-Alveolate-Rhizaria (SAR) group. Panel (**A**) shows a reconstruction of the evolutionary history of candidate PI-PLC homologues inferred from Maximum-Likelihood analyses, using the JTT model [69] with 500 bootstrap replicates [70]. Branches that were reproduced in less than 50% of the bootstrap replicates were collapsed (node information indicates the % of replicates). This analysis used 53 amino acid sequence, with 3274 positions in the final dataset, and was conducted using MEGA11 software [71]. Letters in boxes represent taxonomic groups: A, alveolates; D, discobans; O, oomycetes; R, rhizarians; and T, thraustochytrids. For species names, see Appendix A. Numbers within circles represent protein domain architectures. Panel (**B**) shows these different types of architectures, which are shown approximately to scale. Additionally, shown is a key to protein domains.

**Figure 4 microorganisms-10-02157-f004:**
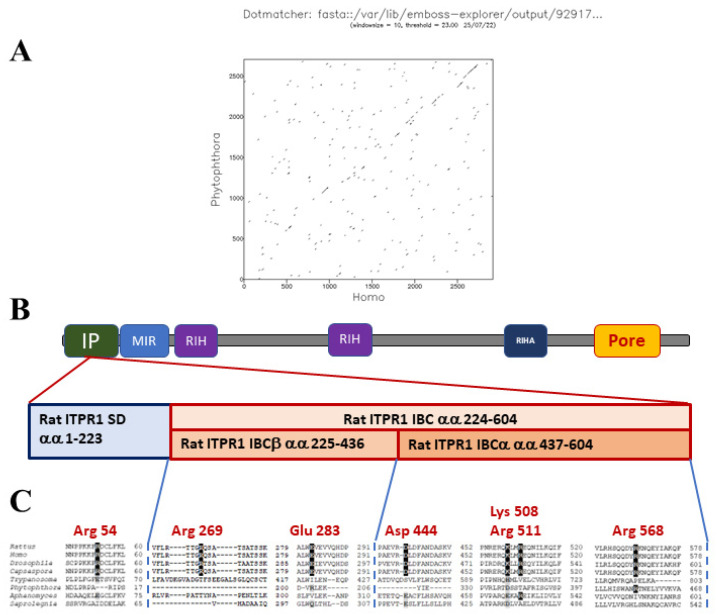
Comparison of IP_3_R proteins from mammals with that from *P. infestans*. Panel (**A**) shows an amino acid identity plot between IP_3_Rs from *H. sapiens* (ITPR1) with that from *P. infestans*. Note the limited identity between the proteins, apart from at the C-terminal channel domain [19]. Panel (**B**) displays the key protein domains (IP: IP_3_-binding domain; MIR: mannosyltransferase, IP_3_R and RyR domain; RIH: RyR and IP_3_R homology domain; RIHA: RIH-associated domain; Pore) within a mammalian IP_3_R (*Rattus norvegicus* ITPR1) and expanded view of the IP domain. This includes the suppressor domain (SD) and the β- and α-segments of the IP_3_ binding core (IBC). Panel (**C**) represents a multiple sequence alignment of candidate IP domain from the IP_3_Rs of several eukaryotic species: *Rattus norvegicus* ITPR1, *Homo sapiens* ITPR1, *Drosophila melanogaster* ITPR, *Capsaspora owczarzaki* ITPR, *Phytophthora infestans* ITPR, *Aphanomyces astaci* ITPR and *Saprolegnia parasitica* ITPR. Note the lack of conservation of residues that are critical for interactions between IP_3_ and mammalian IP domains, indicated in red text.

**Table 1 microorganisms-10-02157-t001:** Small molecules used to investigate PI signalling in oomycetes. This indicates the structural class, targets in mammalian cells and examples of effects on oomycetes, with accompanying reference citations.

Small Molecule	Structural Class	Targets in Mammals	Example of Effects on Oomycetes
U73122	Aminosteroid	PI-PLC inhibitor. Activator of *H.sapiens* PLCβ3. Ca^2+^-pump inhibitor. 5-lipoxygenase inhibitor.	Block of zoosporogenesis in *P.infestans* [41] Inhibition of LIM-interactor transcription factor expression in *P. infestans* [42]
Mastoparan	Peptide	PI-PLC activator. PI-PLC inhibitor. Membrane permeabilization.	Accumulation of phosphatidic acid in *P.infestans* (activation of phospholipase D?) [43]
2-amino- ethoxydiphenyl borate (2-APB)	Diphenylborane	IP_3_R inhibitor SOCE modulator TRP channel activator Ca^2+^-pump inhibitor	Block of zoosporogenesis in *P.infestans* [41]. Inhibition of LIM-interactor transcription factor expression in *P.infestans* [42]
Ethylene glycol-bis(2-amino-ethylether)-N, N, N, N -tetraacetic acid (EGTA)	Aminopolycarboxylic acid	Ca^2+^-chelator Mg^2+^-chelator	Block of cyst germination in *Pythium porphyrae* [39]

## Data Availability

All data used in this review are either presented in this work or in the associated Appendix A.

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
