# Peer review of "Myo-D-inositol Trisphosphate Signalling in Oomycetes"

_microorganisms, 2022, doi:10.3390/microorganisms10112157_

Round 1

Reviewer 1 Report

In the manuscript, authors presents the importance and special aspects of myo-inositol trisphosphate signalling in oomycetes. In most parts of the manuscript, the text is quite good and informative, but at other parts the text is too compact making it difficult to read. However, figures are the weak point of the manuscript and to make it more informative, those could be corrected. I list some comments below, especially related to figures to improve the manuscript.

1. There are some figures already in the manuscript but the number could be increased and also the resolution, legend, and linking to the main text could be improved. Now the resolution in figures is lacking and there are parts that are difficult to understand even with the legend (like figure 1 upper part and figure 2B). Figure legends are quite long but still information for the reader is quite small. Also figures should be more linked to the main text. This would help the reader significantly.

2. There could be more figures to help the reader. For example, simplified signaling figure at the beginning of part 3. Similarly the are multiple inhibitors mentioned so the structure of those and their target could be informative. Same goes with the reactions discussed at the beginning of part 4. And finally at the end there could be figure or table to draw a summary about similarities and differences.

3. Some part are quite hard to read due to the too compact presentation or other reasons. Examples for this are Page 6 lines 274-288 and Page 9 lines 362-364. Authors could try to loosen the presentation and additional figures could also help on this. Also the figure 2A could be earlier in the text so that the reader should be more easily guided through the story.

4. The end and especially the perspective should be deepen to give the reader better overall view. It also should discuss what is missing so what info is lacking and what is ongoing and so on. Also there should be discussion about the differences and how those can be utilized. Is the e.g.  some vulnerability to use for new pesticide development etc.

Author Response

We thank the reviewer for their insightful comments, which have contributed to an improved manuscript. Revisions in response to these useful suggestions are:

1) We have added an additional Table (Table 1, summarizing some of the features of small molecules used to investigate PI signalling) and Figure 1 (an overview of PI signalling in eukaryotes). Descriptions of figures in the text have been revised accordingly. We anticipate that these will improve the clarity and meaning of our review. In terms of the resolution of Figures, we suspect that the versions included in the draft manuscript are of lower resolution than the final versions will be. We will work with the Journal to ensure that this is the case. 

2) Figure 2 summarizes the similarities and differences in PI signalling between oomycetes and other eukaryotes.

3) We have decreased the density of the text throughout, including the specific sections indicated. For example, we have broken long sentences into shorter ones and have simplified some of the phrasing.

4) The Perspectives section has been modified, to include suggestions for future experimentation.

Reviewer 2 Report

the manuscript could be accepted after minor correction, involved with the massage 

Author Response

We would like to thank the reviewer for their useful suggestions, which have been incorporated into a revised manuscript. We have incorporated all of the corrections which the reviewer suggested, which the exception of:

Line 240, change "nor" to "or". The syntax of this sentence is more accurate using the word "nor".